# Levels of mother-to-child HIV transmission knowledge and associated factors among reproductive-age women in Ethiopia: Analysis of 2016 Ethiopian Demographic and Health Survey Data

**Mamo Nigatu Gebre** [1] *, **Merga Belina Feyasa** [2], **Teshome Kabeta Dadi** [1]

1 Department of Epidemiology, Faculty of Public Health, Institute of Health, Jimma University, Jimma, Oromia, Ethiopia, 2 Department of Statistics, College of Natural & Computational Sciences, Addis Ababa University, Addis Ababa, Ethiopia

* mamogebre14@gmail.com

**Data Availability Statement:** All data and materials used in this study are openly accessed and

## Abstract

### Background

The world community has committed to eliminating the mother-to-child transmission of human immunodeficiency virus. Even though different studies have been done in Ethiopia, to the knowledge of the investigators, the Ethiopian women's level of knowledge on the mother-to-child transmission of human immunodeficiency virus is not well studied and the existing evidence is inconclusive. The current study is aimed to study the Ethiopian women's level of knowledge on the mother-to-child transmission of human immunodeficiency virus and its associated factors using the 2016 Ethiopian Demographic and Health Survey Data.

### Methods

Data of 15,683 women were extracted from the 2016 Ethiopia Demographic and Health Survey. Descriptive statistics and multilevel ordinal logistic regression were respectively used for the descriptive and analytical studies.

### Results

41.1% [95% CI: 39.5%, 42.7%] of the Ethiopian reproductive-age women have adequate knowledge of the mother-to-child transmission of human immunodeficiency virus. 77%, 84% and 87.8% of the women respectively know that human immunodeficiency virus can be transmitted during pregnancy, delivery, and breastfeeding. There are wider regional variations in the women's level of knowledge of the mother-to-child transmission of human immunodeficiency virus. Being an urban resident, having better educational status, being from a wealthy household, owning of mobile phone, frequency of listening to the radio, frequency of watching television, and being visited with field workers were significantly

available on a public domain MEASUREDHS website. URLs:https://dhsprogram.com/data/.

**Funding:** The author(s) received no specific funding for this work.

**Competing interests:** The authors have declared that no competing interests exist.

**Abbreviations:** AIC, Akaike Information Criterion; AIDS, Acquired immune deficiency syndrome; AOR, Adjusted Odds Ration; BIC, Bayesian Information Criterion; CI, Confidence Interval; CSA, Central Statistical Agency; DHS, Demographic and Health Survey; EA, Enumeration Area; EDHS, Ethiopian Demographic and Health Survey; HIV, Human Immunodeficiency Virus; ICC, Intraclass correlation; MTCT, Mother-To-Child Transmission; PHC, Population and Housing Census; PMTCT, Prevention of Mother-To-Child Transmission; PPS, Probability Proportionate to Size; UNAIDS, United Nations Program on HIV/AIDS; UNICEF, United Nations Children's Fund; VCT, Voluntary Counseling and Testing; WHO, World Health Organization.

associated with having adequate knowledge of the mother-to-child transmission of human immunodeficiency virus.

## Conclusion

Despite all collective measures put in a place by different stakeholders to prevent the mother-to-child transmission of HIV in Ethiopia, a large proportion of the Ethiopian women do not know about the mother-to-child transmission of the disease. Stakeholders working on HIV prevention and control should give due emphasis to promoting mobile phone technology and other media like radio and television by giving due focus to rural residents and poor women to promote the current low level of the knowledge. Emphasis should also be given to the information, education, and communication of the mother-to-child transmission of the disease through community-based educations.

## Background

Since the beginning of the pandemic, Human immunodeficiency Virus (HIV) has infected more than 75 million people and claimed about 32 million lives. Globally, 1.7 million people were newly infected and about 37.9 million people were living with HIV at the end of 2018 [1–3]. Sub-Saharan Africa disproportionately carries a highest burden of HIV accounting for more than 70% of the global burden of the infection. More two-third of the estimated 6000 new HIV infections that occur globally each day occurs in sub-Saharan Africa where young women disproportionately bear a highest burden of the disease. East and Southern Africa is the most affected region in the world and is home to the largest number of people living with HIV [4, 5]. In Ethiopia, according to the 2016 WHO report, an estimated number of 710,000 were living with HIV; however, according to the 2019 UNAIDS report, the number was decreased to 690,000 at the end of 2018 [6, 7]. In many countries where HIV is prevalent, women continue to acquire HIV during pregnancy and breastfeeding and risk transmitting the disease to their infants [8]. The transmission of HIV from an HIV-positive mother to her child during pregnancy, labor, delivery, or breastfeeding is called mother-to-child transmission (MTCT). The rate of transmission of HIV from an HIV-positive mother to her baby ranges from 15 percent to 45 percent in the absence of intervention and can be reduced to below 5 percent with an effective intervention [9]. Child born from HIV mother is at risk of contracting the disease [10, 11]. The MTCT of HIV, which is also known as 'vertical transmission', accounts for the majority of infections in children aged 0–14 years [8, 12]. A systematic review and meta-analysis done using 33 published article depicted that the pooled prevalence of MTCT of HIV in the East Africa is 7.68% [11].

The global community has committed to eliminating MTCT of HIV as a public health priority through a harmonized and integrated approach to improve the health outcomes of mothers and their children [13, 14]. The Joint United Nations Program on HIV/AIDS had launched a global plan in 2011 which covered all low-and middle-income countries with due focuses on 22 countries where 90% of all pregnant women living with HIV reside to eliminate new HIV infections among children by 2015 and keeping their mothers alive. The program had also adopted a new strategy in October 2015 to end the AIDS epidemic as a public health threat by 2030 with an interim goal of 95% coverage with antiretroviral therapy among pregnant women and less than 20 000 new pediatric HIV infections by 2020, and gained tremendous

achievements [13–16]. According to the 2016 UNAIDS report, new HIV infections among children were reduced by 60% in 21 countries which were badly hit by the disease in Sub-Saharan Africa (SSA), and 6 countries had cut new infections among children by 75% or more [17]. On the other hand, new HIV infection among children aged 0–14 years was reduced by about 41% in 2018 as compared to the new infection in 2010 [18].

However, even though there is promising progress in the HIV response, the existing pieces of evidence have depicted that the disease is still devastating the lives of many children. The 2020 UNICEF report showed that, of the estimated 38.0 million people living with HIV worldwide in 2019, 2.8 million were children aged 0–19 years. The same report showed that approximately 880 and 310 children became infected with and died from AIDS-related causes respectively on each day in 2019, mostly because of inadequate access to HIV prevention, care, and treatment services [19]. The study in Belgaum district, Karnataka, India depicted that the HIV prevalence rate among babies exposed to maternal HIV until 24 months was 7.8% [20]. The 2017 WHO report showed that an estimated number of 62000 children aged 0–14 years were living with HIV in Ethiopia in 2016 which accounted for 8.7% of the total infection [6]. The systematic review and meta-analysis from Ethiopia also revealed that the pooled prevalence of MTCT of HIV in Ethiopia was 9.93% [21]. Similarly, the retrospective cohort study done at Northwest Ethiopia depicted that the prevalence of MTCT of HIV among HIV-exposed infants was 10 percent [22].

Prevention of mother-to-child transmission (PMTCT) programs offer a myriad of services for reproductive-age women living with HIV or at risk of the infection to maintain their health and prevent their infants from contracting the disease. A woman in a reproductive-age group should be offered the PMTCT services before conceiving a child, throughout the entire pregnancy, labor, and breastfeeding. The PMTCT services had prevented nearly 1.4 million new HIV infections among children between 2010 and 2018 [8, 12, 13]. In Ethiopia, the PMTCT services are given integrated with other routine, maternal and child health care services with the guiding principles of equity, human right, integration, family focused, prioritizing pregnant women with advanced disease for HAART, standardization, referral and linkage, confidentiality and voluntary informed consent, community participation and male partner involvement. The country had adopted the 4-pronged WHO/UNICEF/UNAIDS PMTCT strategy: primary prevention of HIV infection, prevention of unintended pregnancy among HIV infected women, prevention of HIV transmission from infected women to their infants and Treatment, care and support of HIV infected women, their infants and their families as a key entry point to HIV care for women, men and their infants [23, 24]. Even though the PMTCT of HIV services were proven effective in preventing the vertical transmission of HIV from mother-to-child, it was evidenced that a large proportion of reproductive-age women do not know about vertical transmission of HIV. A study from Durban; South Africa showed that the majority of the reproductive-age women do not know MTCT of HIV. The same study concluded that more innovative ways to impart knowledge particularly of PMTCT and updated standards of practice are needed [25]. The different studies done in the different regions of Ethiopia have evinced that 34.5%-81% of the reproductive-age women residing in the country do not have knowledge of MTCT of HIV [26–31]. The studies have also shown that urban residence, secondary and above educational level, attending antenatal care, receiving information from health care providers, wealth status, and exposure to mass media were among the common factors associated with women's knowledge of MTCT of HIV [26, 27, 29, 30].

Ethiopia is a landlocked country bordering Eritrea, Somalia, Kenya, South Sudan, and Sudan. The country has been using neighboring Djibouti's main port for the last two decades, however, with the recent peace agreement with Eritrea, it is set to resume accessing the Eritrean ports of Assab and Massawa for its international trade. With a population of more than

112 million, Ethiopia is the second most populous nation in Africa following Nigeria. The country is among the fast-growing economy in the region; however, it is also one of the poorest, with a per capita income of $850. Administratively, the country is divided into nine regions and two city administrations [32, 33]. Even though different small-scale studies have been done in Ethiopia to study women's level of knowledge on MTCT of HIV and its associated factors, to the best knowledge of the investigators, the problem is not well studied using nationally representative data, and the existing evidence is inconclusive. Hence, the current study is aimed to study the level of mother-to-child HIV transmission knowledge and its associated factors among reproductive-age women in Ethiopia using the nationally representative 2016 Ethiopian Demographic and Health Survey (EDHS) Data.

## Materials and methods

### Data sources

The 2016 EDHS data were collected by the Central Statistical Agency (CSA) and other stakeholders both in Ethiopia and abroad. The authors accessed the processed and organized data from open datasets of the MEASUREDHS by permission. Variables anticipated to be associated with women's knowledge about mother-to-child HIV transmission were extracted from the 'women dataset' based on the reviewed literature and then processed for further analyses. For collecting the EDHS data, standard protocols and three types of tools; the Household Questionnaire, the Woman's Questionnaire, and the Man's Questionnaire were used. Further contextualization and standardization of the questionnaires were also done by governmental and non-governmental shareholders to maintain the validity of the tools [33].

### Study population and sampling procedures for the 2016 EDHS

The 9 regions and 2 two city administrations in Ethiopia were considered based on the 2007 census that divided each kebele, the lowest governmental administration unit, to a subdivision called census enumeration areas (EAs). The survey followed a two-stage sampling design with stratification into urban and rural. At the first stage, 645 EAs, 202 from urban, and 443 from a rural were selected according to probability proportionate to the size of the EAs. At the second stage, approximately 28 households from each EA were selected by systematic random sampling and then all women whose ages were from 15–49 who live in the selected households were included in the study [34]. By this procedure, 15,683 eligible women were identified and interviewed during the parent study and so were considered for the current study.

### Measurements

The dependent variable of the study was knowledge of mother-to-child HIV transmission among reproductive-age women and generated from the scoring of four questions each woman was asked. The questions were "the virus that causes AIDS can be transmitted from a mother to her baby during pregnancy"; "The virus that causes AIDS can be transmitted from a mother to her baby during delivery"; "The virus that causes AIDS can be transmitted from a mother to her baby by breastfeeding"; and "There are special drugs that a doctor or a nurse can give to a woman infected with the AIDS virus to reduce the risk of transmission to the baby" [33, 35]. The scoring which ranges from '0' to '4' was done from the responses of the 4 questions and then grouped into 3 levels of knowledge of MTCT of HIV.

## Operational definitions

**No knowledge of mother-to-child transmission.** If the score of the four measurement questions of MTCT of HIV summed to '0'.

**Inadequate knowledge of mother-to-child transmission.** If the scores of the four measurement questions of MTCT of HIV sum ranges from 1 to 3.

**Adequate knowledge of mother-to-child transmission.** If the scores of the four measurement questions of MTCT of HIV summed to '4' i.e. if they answered the four questions correctly.

**Agricultural workers.** Refers to those females who were market-oriented skilled agricultural workers, market-oriented skilled forestry, fishery and hunting, and agricultural, forestry, and fishery laborers.

**Professional workers.** Include chief executives, senior officials, and legislators, administrative and commercial managers, production and specialized services managers, science and engineering professionals, health professionals, teaching professionals, and other professionals were included.

**Trade or sales workers.** Encloses sales workers, building and related trades workers, excluding electricians, metal, machinery and related traders, handicraft and printing workers and, electrical and electronic traders.

**Elementary occupation.** Covers cleaners and helpers, laborers in mining, construction, manufacturing and transport, food preparation assistants, street and related sales and service workers and, refuse workers and other elementary workers.

**Others workers.** Consists of hospitality, retail, and other services managers, general and keyboard clerks, numerical and material recording clerks, other clerical support workers, personal service workers, personal care workers, protective services workers, handicraft and printing workers, food processing, woodworking, garment, and other craft and related trades workers.

## Data analysis

Data processing, management, and analyses were performed using Stata 14.2 statistical software. The multilevel ordinal logistic regression model was fitted to assess regional variation of knowledge of women about MTCT of HIV and to identify factors associated with the outcome of interest for the target population of reproductive-age women in Ethiopian. The appropriate statistical method that can capture inflation of variability due to the application of staged sampling is multilevel analysis. Models used for the analysis of hierarchical data structure must account for associations among observations within clusters to make efficient and valid inferences. When the variance of the residual errors is correlated between individual observations as a result of these nested structures, single ordinal logistic regression is inappropriate [36]. Consequently, in this study, multilevel ordinal logistic regression was used to assess the relationship between levels of knowledge of HIV and associated factors using the 2016 EDHS data.

In the analysis of multilevel regression, the clustering effect plays a great role in the estimation of the parameters and this clustering effect can be quantified by intraclass correlation (ICC). ICC is the proportion of total variation in the response variable that is accounted for by between-group variation [36]. In this study, the effect of the clustering variable (region) where the subjects were residing during the study period was given an emphasis and all other predictors were considered at the first level.

All the outputs for descriptive as well as fitting multilevel ordinal logistic regression analyses were carried out by weighting provided by the MEASUREDHS program. The weights from DHS were used to carry out multilevel analysis but adjusted as per the recommendation by

Adam [37]. Moreover, we have checked the goodness of fit after weighting the dataset by both the DHS and Adam's. Compared to the multilevel ordinal logistic regression fitted by using the unadjusted weights (AIC = 30,307.7, BIC = 30,392.0), the model fitted using the adjusted weights (AIC = 25,826.1, BIC = 25,902.7) had lower AIC and BIC. Besides the choice of weights, the model with fewer numbers of variables in the model was considered due to the principle of parsimony. The final model with significant variables is presented in Table 3.

### Ethics statement

The EDHS 2016 survey protocol was reviewed and approved by the Federal Democratic Republic of Ethiopia, Ministry of Science and Technology, and the Institutional Review Board of ICF International. Additionally, written consent was obtained from each respondent. All participant identifiers were removed during data entry of the parent study, earlier of doing data management and any analyses. For the current study, the authors received permission from the public domain MEASUREDHS website and re-analyzed the data.

## Results

### Characteristics of the participants

Fully, 15,683 participants were included in this study and the average age of the participants was 28.17 (± 9.16) year. On average, each woman had 0.7 (±0.84) number of births in the last five years and had 2.37(±2.34) mean number of antenatal visits during pregnancy. As presented in Table 1, regarding religion, 43.3%, 23.4%, and 31.2% of the participants were Orthodox, Protestant, and Muslim religious followers respectively. Most of the study participants (65.2%) were in a union by the time the survey was conducted, whereas about one in four were never been in a union. Half (50%) of the women were not engaged in a paid type of work and 20.8% were engaged in agricultural works. The majority of the participants (77.8%) were rural residents. Of all women included in this study, 47.8% never attended school; 35.0%, 11.6%, and 5.6% attended primary, secondary and higher education, respectively. More than 50% of the participants belong to the poor to the middle class of wealth status. Concerning the partners of the respondents, about 84% of them attended primary education; whereas most of the partners close to 63% were engaged in agricultural works.

### Antenatal care-related and individual characteristics of reproductive-age women in Ethiopia, EDHS 2016

About 46.3% of women had the opportunity to talk about the transmission of HIV from mother-to-child during the antenatal visit; whereas 47% of them also discussed how to prevent HIV. More than half (59.0%) of the participants in this study were tested for HIV as a part of the antenatal care during the visits. Government health centers were the dominant (70.7%) places where HIV tests were given as part of the antenatal visit. Nearly half (48.4%) of the women had at least one birth in the past five years before the survey. Almost four in nine of the women were tested for HIV; in contrast one in four of them do not know where to get tested for HIV. On the other hand, only 36% of the women discussed with health workers about family planning. Often (66%) decisions about health care were made by both respondent and husband/partner. About two in five of the women visited a health facility in the last 12 months before the survey. Only 7.2% of the women were pregnant by the time the survey was conducted (Table 2).

**Table 1. Socio-demographic and economic characteristics of reproductive-age women in Ethiopia, EDHS 2016.**

| Variables | Categories | Total (%) | Knowledge of MTC HIV transmission | | | | | |
|---|---|---|---|---|---|---|---|---|
| | | | No | | Inadequate | | Adequate | |
| | | | N | % | N | % | N | % |
| **Religion** | Orthodox | 6,786(43.3) | 814 | 12.0 | 2,784 | 41.0 | 3,188 | 47.0 |
| | Protestant | 3,674(23.4) | 723 | 19.7 | 1,701 | 46.3 | 1,250 | 34.0 |
| | Muslim | 4,893(31.2) | 1,389 | 28.4 | 2,103 | 43.0 | 1,400 | 28.6 |
| | Other | 330(2.10) | 84 | 25.5 | 149 | 45.2 | 97 | 29.3 |
| **Marital Status of respondents** | Never in union | 4,036(25.7) | 627 | 15.5 | 1,767 | 43.8 | 1,642 | 40.7 |
| | Currently in union | 10,223(65.2) | 2,114 | 20.7 | 4,436 | 43.4 | 3,674 | 36.0 |
| | Formerly in union | 1,423(9.1) | 269 | 18.9 | 535 | 37.6 | 619 | 43.5 |
| **Occupational Status** | Not working | 7,819(49.9) | 1,762 | 22.5 | 3,348 | 42.8 | 2,709 | 34.7 |
| | Agricultural Workers | 3,263(20.8) | 613 | 18.8 | 1,534 | 47.0 | 1,116 | 34.2 |
| | Professionals | 390(2.5) | 22 | 5.5 | 114 | 29.2 | 255 | 65.3 |
| | Trade/Sales | 2,070(13.2) | 288 | 13.5 | 912 | 44.1 | 870 | 42.0 |
| | Elementary work | 1,070(6.8) | 144 | 13.5 | 409 | 38.2 | 468 | 43.7 |
| | Others | 1,069(6.8) | 181 | 16.9 | 421 | 39.4 | 468 | 43.7 |
| **Residence** | Urban | 3,476(22.2) | 281 | 8.1 | 1,125 | 32.4 | 2,070 | 59.6 |
| | Rural | 12,207(77.8) | 2,730 | 22.4 | 5,613 | 46.0 | 3,864 | 31.7 |
| **Respondents educational level** | No education | 7,498(47.8) | 2,104 | 28.1 | 3,227 | 43.0 | 2,167 | 28.9 |
| | Primary | 5,490(35.0) | 767 | 14.0 | 2,568 | 46.8 | 2,156 | 39.3 |
| | Secondary | 1,817(11.6) | 111 | 6.1 | 680 | 37.4 | 1,026 | 56.4 |
| | Higher | 877(5.6) | 29 | 3.3 | 263 | 30.0 | 586 | 66.8 |
| **Wealth Status** | Poor | 5,442(34.7) | 1,576 | 29.0 | 2,409 | 44.3 | 1,457 | 26.8 |
| | Middle | 2,978(19.0) | 601 | 20.2 | 1,419 | 47.7 | 958 | 32.2 |
| | Rich | 7,263(46.3) | 833 | 11.5 | 2,910 | 40.1 | 3,520 | 48.5 |
| **Partner educational level** | No education | 4,685(46.2) | 1,247 | 26.6 | 2,069 | 44.2 | 1,368 | 29.2 |
| | Primary | 3,772(37.2) | 725 | 19.2 | 1,695 | 44.9 | 1,352 | 35.8 |
| | Secondary | 975(9.6) | 91 | 9.3 | 400 | 41.0 | 485 | 49.7 |
| | Higher | 713(7.0) | 2099 | 20.7 | 4391 | 43.3 | 3655 | 36.0 |
| **Occupational Status of the partner** | Not working | 807(8.0) | 306 | 37.9 | 305 | 37.8 | 196 | 24.2 |
| | Agricultural Workers | 6327(62.6) | 1,356 | 21.4 | 3,000 | 47.4 | 1,970 | 31.1 |
| | Professionals | 754(7.5) | 66 | 8.8 | 253 | 33.5 | 435 | 57.7 |
| | Trade/Sales | 1075(10.6) | 164 | 15.3 | 435 | 40.5 | 475 | 44.2 |
| | Elementary work | 665(6.6) | 108 | 16.2 | 249 | 37.5 | 308 | 46.3 |
| | Others | 475(4.7) | 69 | 14.7 | 154 | 32.3 | 252 | 53.0 |

## Level of mother-to-child HIV transmission knowledge of reproductive-age women in Ethiopia

The results of this study showed that only 41.1% [95% CI: 39.5%, 42.7%] of the Ethiopian reproductive-age women have adequate knowledge of MTCT of HIV/AIDS, whereas, 39.0% [95% CI: 37.7%, 40.3%] and 19.9% [95% CI: 18.4%, 21.5%] of the women have inadequate and no knowledge of the MTCT of HIV respectively. The study also revealed that 77%, 84%, and 87.8% of the women respectively know that HIV can be transmitted from a mother to her child during pregnancy, delivery, and breastfeeding (Fig 1). There are wider regional disparities in the level of the knowledge of the MTCT of HIV among the women residing in the different regions of the country. Nearly two-thirds (66.8%) of the women residing in the Addis Ababa region have an adequate knowledge of the MTCT of HIV followed by the women

**Table 2. Antenatal care-related and individual characteristics of reproductive-age women in Ethiopia, EDHS 2016.**

| Variables | Categories | Total (%) | Knowledge of MTC HIV transmission | | | | | |
|---|---|---|---|---|---|---|---|---|
| | | | No | | Inadequate | | Adequate | |
| | | | N | % | N | % | N | % |
| During antenatal visit talked about: HIV transmitted mother to child | No | 1,660(53.7) | 312 | 18.8 | 894 | 53.9 | 454 | 27.3 |
| | Yes | 1,431(46.3) | 74 | 5.2 | 491 | 34.3 | 865 | 60.5 |
| During antenatal visit discussed how to prevent HIV | No | 1,648(53.0) | 325 | 19.7 | 873 | 52.9 | 451 | 27.4 |
| | Yes | 1,462(47.0) | 76 | 5.2 | 520 | 35.6 | 866 | 59.3 |
| During antenatal visit talked about: getting tested for HIV | No | 1333(42.7) | 291 | 21.9 | 704 | 52.9 | 337 | 25.3 |
| | Yes | 1789(57.3) | 112 | 6.3 | 985 | 38.6 | 985 | 55.1 |
| Offered HIV test as part of antenatal visit | No | 1,304(41.5) | 248 | 19.0 | 716 | 54.9 | 341 | 26.1 |
| | Yes | 1,840(58.5) | 168 | 13.2 | 1,403 | 44.6 | 1,325 | 42.2 |
| Tested for HIV as part of antenatal visit | No | 1,289(41.0) | 262 | 20.3 | 709 | 55.0 | 3,182 | 24.5 |
| | Yes | 1,855(59) | 154 | 8.3 | 694 | 37.4 | 1,007 | 54.3 |
| The place where the HIV test was taken as part of the antenatal visit | Government Hospital | 211(11.4) | 13 | 6.1 | 74 | 35.2 | 124 | 58.7 |
| | Government health center | 1312(70.7) | 110 | 8.4 | 497 | 37.9 | 705 | 53.7 |
| | Government health post | 236(12.7) | 26 | 11.2 | 95 | 40.2 | 115 | 48.6 |
| | Other public sector | 96(5.2) | 5 | 5.4 | 27 | 28.2 | 64 | 66.4 |
| Births in the last five years | No birth | 8,093(51.6) | 1,342 | 16.6 | 3,481 | 43.0 | 3270 | 40.4 |
| | $\geq$ 1 birth | 7590(48.4) | 1,668 | 22.0 | 3,257 | 42.9 | 2,664 | 35.1 |
| Ever been tested for HIV | No | 8,753(55.8) | 2,450 | 28.0 | 3,980 | 45.5 | 2,323 | 26.6 |
| | Yes | 6,930(44.2) | 561 | 8.1 | 2758 | 39.8 | 3611 | 52.1 |
| Know a place to get an HIV test | No | 3,723(25.5) | 827 | 22.2 | 2,095 | 56.3 | 801 | 21.5 |
| | Yes | 10,876(74.5) | 1,099 | 10.1 | 4,643 | 42.7 | 5,133 | 47.2 |
| Discussion with HW about FP | No | 4,176(64.0) | 635 | 15.2 | 1,812 | 43.4 | 1,729 | 41.4 |
| | Yes | 2,350(36) | 255 | 10.9 | 912 | 38.8 | 1183 | 50.4 |
| A decision about Health care | Respondent alone | 1575(15.4) | 340 | 21.6 | 631 | 40.0 | 604 | 38.4 |
| | Respondent and husband/partner | 6749(66.0) | 1,281 | 19.0 | 2,845 | 42.2 | 2,623 | 38.9 |
| | husband/partner alone | 1,858(18.2) | 478 | 25.7 | 942 | 50.7 | 438 | 23.6 |
| | someone else | 42(0.4) | 14 | 34.7 | 18 | 43.8 | 9 | 21.6 |
| Visited health facility last 12 months | No | 9,157(58.4) | 2,120 | 23.2 | 4,014 | 43.8 | 3,022 | 33.0 |
| | Yes | 6,526(41.6) | 890 | 13.6 | 2,724 | 41.7 | 2,912 | 44.6 |
| Current pregnancy status | No | 14548(92.8) | 2782 | 19 | 6221 | 42.8 | 5545 | 38.1 |
| | Yes | 1135(7.2) | 229 | 20.1 | 517 | 45.5 | 390 | 34.3 |

residing in the Tigray (50.7%) and Harari (47.6%) regions. The women residing in the Somali region are the least to know MTCT of HIV where only 11.3% of the women have adequate knowledge and 60.5% of the women do not have any knowledge of the MTCT of HIV (Fig 2). The multilevel ordinal logistic regression also revealed that about 13.3% of the variation in the level of women's knowledge of the MTCT of HIV was explained by the variations among the regions.

## Factors associated with mother-to-child HIV transmission knowledge among reproductive-age women in Ethiopia, EDHS 2016

In all the forthcoming interpretations, by odds ratio, we mean adjusted odds ratio (AOR) and while interpreting AOR for a selected variable we further assume that all the other variables in the model are held constant.

The residence of the respondents has a statistically significant association with the level of knowledge of mother to child HIV transmission. The odds of having adequate knowledge of

**Table 3. Factors associated with mother-to-child HIV transmission knowledge among reproductive-age women in Ethiopia, EDHS 2016.**

| Knowledge of mother to child HIV transmission | Odds Ratio | Robust Std. Err. | z | P>z | [95% Conf. Int.] | |
|---|---|---|---|---|---|---|
| **Residence** (Ref. Urban) | 1 | | | | | |
| Rural | 0.71 | 0.07 | -3.37 | 0.001 | 0.58 | 0.86 |
| **Education** (Ref. No education) | 1 | | | | | |
| Primary | 1.47 | 0.09 | 6.19 | 0.000 | 1.30 | 1.65 |
| Secondary | 1.71 | 0.18 | 5.10 | 0.000 | 1.39 | 2.09 |
| Higher | 1.81 | 0.24 | 4.48 | 0.000 | 1.40 | 2.34 |
| **Wealth Status** (Ref. Poor) | 1 | | | | | |
| Middle | 1.33 | 0.11 | 3.48 | 0.001 | 1.13 | 1.56 |
| Rich | 1.52 | 0.11 | 5.70 | 0.000 | 1.32 | 1.76 |
| **Owns Mobile** (Ref. No) | | | | | | |
| Yes | 1.43 | 0.07 | 7.66 | 0.000 | 1.30 | 1.56 |
| **Freq. of listening Radio** (Ref. Not at all) | 1 | | | | | |
| Less than 1 a week | 1.24 | 0.06 | 4.58 | 0.000 | 1.13 | 1.36 |
| At least once a week | 1.27 | 0.06 | 5.33 | 0.000 | 1.16 | 1.39 |
| **Freq. of watching TV** (Ref. Not at all) | 1 | | | | | |
| Less than 1 week | 1.28 | 0.09 | 3.42 | 0.001 | 1.11 | 1.47 |
| At least once a week | 1.61 | 0.14 | 5.70 | 0.000 | 1.37 | 1.90 |
| Visited by fieldworker in the last 12 months (Ref. No) | 1 | | | | | |
| Yes | 1.38 | 0.08 | 5.44 | 0.000 | 1.23 | 1.55 |
| /cut1 | | -0.84 | 0.26 | -3.23 | 0.001 | -1.35 | -0.33 |
| /cut2 | | 1.27 | 0.22 | 5.73 | 0.000 | 0.84 | 1.70 |
| **Region** | | | | | | |
| var(_cons) | 0.26 | 0.16 | | 0.076 | 0.90 | |

mother-to-child HIV transmission instead of none or inadequate for rural dwellers was lower by approximately 30%; AOR = 0.71 with a 95% CI [0.58, 0.86] as compared to the urban dwellers. Education has a statistically significant association with the level of knowledge of the mother-to-child HIV transmission. The odds of having an adequate level of knowledge for primary, secondary and high-schoolers were AOR = 1.47 with a 95% CI [1.3, 1.65], AOR = 1.71 with a 95% CI [1.39, 2.09] and AOR = 1.81 with a 95% CI [1.40, 2.34] respectively compared to women who have never been to school. The participants with middle income and the riches were more likely to have an adequate level of knowledge of mother-to-child HIV transmission, AOR 1.33 with a 95% CI [1.13, 1.56], and AOR 1.52 with a 95% CI [1.32, 1.56] respectively. The odds of having an adequate level of knowledge of mother-to-child HIV transmission for mobile telephone owners are AOR = 1.43 with a 95% CI [1.30, 1.56] compared to none owners. Both the frequency of listening to a radio and frequency of watching TV were significantly associated with having a higher level of knowledge of mother-to-child HIV transmission. The odds of having an adequate level of knowledge of mother-to-child HIV transmission increase with frequency of listening to a radio with AOR = 1.24, a 95% CI [1.13, 1.36] and AOR = 1.27, a 95% [1.16, 1.39] for those who listen less than once a week and for those listening at least once a week respectively, compared to those who do not listen to the radio at all. The same holds for the frequency of watching TV. The participants who were visited by fieldworkers in the last 12 months before the survey was conducted were more likely to have an adequate level of knowledge of mother-to-child HIV transmission AOR = 1.38 with a 95% CI [1.23, 1.55] (Table 3).

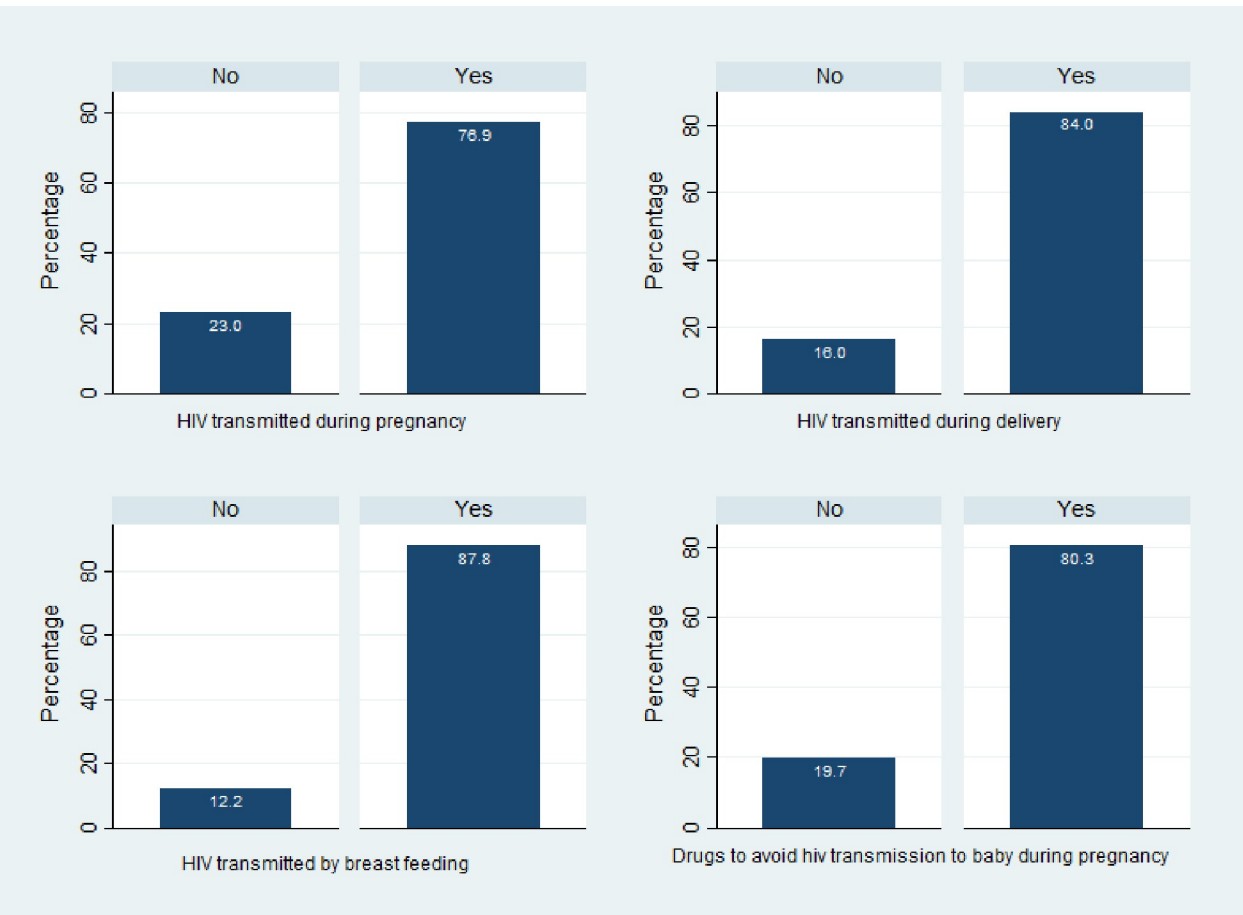

**Fig 1. Percentage of Ethiopian reproductive-age women who know mother-to-child HIV transmission routes, 2016 EDHS.**

## Discussions

The results of this study showed that nearly one-fifth (19.9%) of the reproductive-age women in Ethiopia do not have any knowledge of how HIV is transmitted from mother-to-child, whereas nearly two-fifth (39.0%) of them have inadequate knowledge, and only 41.1% have adequate knowledge. The study also revealed that about 13.3% of the variation in the level of women's knowledge of the MTCT of HIV was explained by the variations among the regions; 66.8% of the reproductive-age women residing in the Addis Ababa region have adequate knowledge of the mother-to-child transmission of HIV, whereas, only 11.3% of the reproductive-age women residing in the Somali region do have the adequate knowledge of the mother-to-child transmission of HIV The current study indicates that Ethiopian women's knowledge of MTCT of HIV has shown little improvement as compared to the findings from the secondary data analysis of the previous EDHS (the 2011 EDHS) during which only 34.9% of the women had adequate knowledge of MTCT of HIV [28]. The finding from the current study is higher than the finding from the study done in Cameroon where only 37% of women had adequate knowledge of MTCT of HIV [38] and the finding from the study done in Northwest Ethiopia where only 19% of the women who had been included in the study knew the MTCT of HIV [26]. However, the finding from the current study is lower than that of the studies done in Zimbabwe and Tanzania where 70.5% and 46% of the reproductive-age women respectively had a comprehensive knowledge of the MTCT of HIV [35, 39] and the two other

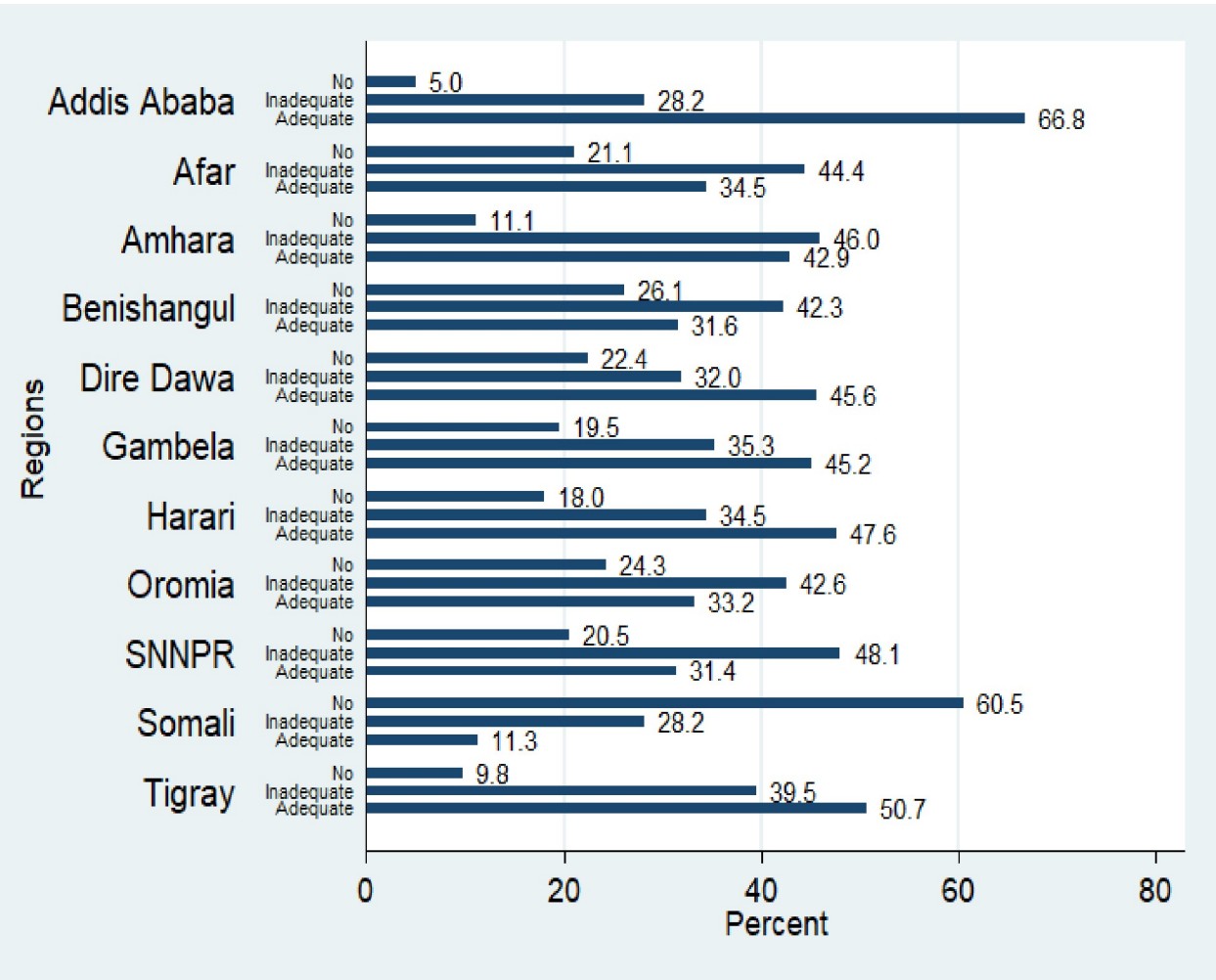

**Fig 2. Levels of mother-to-child HIV transmission knowledge of Ethiopian reproductive-age women by regions in Ethiopia, 2016 EDHS.**

studies done in Southwest Ethiopia and Northern Ethiopia [29, 31] where 65.9% and 52% of the women respectively had a comprehensive knowledge of the MTCT of HIV. The difference might be explained by the differences in uptake of maternal health care services among the reproductive-age women residing in the different regions. The 2016 EDHS report showed that the maternal health care services utilization highly varies across the difference regions of Ethiopia. According to the same report, for example, the ANC coverage from a skilled provider varies highly across the different regions of the country being highest in Addis Ababa (97%) and lowest in Somali (44%) [33]. According to the guideline for prevention of mother-to-child transmission of HIV in Ethiopia, one of the integral components of antenatal care service in the country is routine offer of HIV counselling and testing which might promote the women's knowledge of the mother-to-child transmission of HIV [23, 24].

The finding of the current study also showed that rural-resident women are less likely to have adequate knowledge of MTCT of HIV instead of no knowledge or inadequate knowledge. This finding is in line with the findings from the different small-scale studies done in Ethiopia where urban-resident women had better knowledge of MTCT of HIV [26–28]. This might be because urban-resident women might have better access to maternal health care services and

mass media than rural-resident women. The 2016 EDHS report showed that urban women are more likely than rural women to receive any ANC from a skilled provider; 90% of urban women received any ANC service from a skilled provider as compared to 58% rural women [33]. The study done in Amhara regional state, Ethiopia showed that ANC service utilization has positive significant association with knowledge of the mother-to-child transmission of HIV [40]. The current study and the previous study done in Ethiopia [28] showed that exposure to mass media has a positive and significant association with MTCT of HIV-related knowledge of reproductive-age women. On the other hand, urban-resident women might have better access to education than rural-residents, which might boost their knowledge of MTCT of HIV. The 2016 Ethiopian Demographic and Health Survey report showed that 57% of rural women have no formal education as compared with 16% of urban women [33]. The current study and many other studies [26–30, 38] have shown that a better educational level is positively associated with better MTCT of HIV knowledge among reproductive-age women.

The odds of having adequate knowledge of MTCT of HIV instead of not having this knowledge or having inadequate knowledge among the reproductive-age women who attended primary, secondary and higher education as compared to those who have never attended formal school were 1.47, 1.71, and 1.81 respectively. This finding is supported by the findings from many other studies done in Ethiopia [26–30]. This might also be explained by different factors. Firstly, women may get MTCT of HIV-related knowledge through their formal academic process. On the other hand, women with better academic status might have the ability to gain more MTCT of HIV-related knowledge through their day-to-day life experience as they might have better communication skills. Educated women might also have better access to mass media which has a positive impact on MTCT of HIV-related knowledge among women.

The wealth status of the households in which the women live was significantly associated with knowledge of MTCT of HIV. Those women who were from the rich and the middle-income households were 33% and 52% more likely to have adequate knowledge of MTCT of HIV instead of having no or inadequate knowledge respectively. This finding is concordant with the finding from the study done in Tanzania using a nationally representative sample [35] and the finding from the secondary data analysis of the 2011 EDHS [28] where women from higher wealth quantile households had higher knowledge of MTCT of HIV as compared to women from the lowest wealth quantile households. This could be explained by the inequalities in accessing educational services, health care services, and social media between women from rich households and those from poor households which might have significant impacts on MTCT of HIV-related knowledge among the women. The 2016 Ethiopian Demographic and Health Survey report showed that educational attainment highly varies by wealth quintile; 74% of women in the lowest wealth quintile have no formal education, as compared with 19% of women in the highest wealth quintile. The same report also showed that there is high disparity between rich and poor women in accessing mass media; only 1% of women in the lowest wealth quintile read a newspaper at least once a week, compared with 10% of women in the highest quintile [33]. Different studies had also witnessed that wealth quantile has positive significant association with maternal health care services utilization [41, 42]. The study from Ethiopia using data from the three-round EDHSs (the year 2000, 2005, and 2011) showed that socioeconomic inequality among the reproductive-age women had highly disadvantages the poor women in the uptake of maternal health care services. The same report has shown that inequalities in education and media access significantly contribute to inequalities in maternal health service utilization favoring the non-poor [43].

Owning mobile telephones was also significantly associated with MTCT of HIV-related knowledge among reproductive-age women in Ethiopia. The odds of having adequate knowledge of MTCT of HIV instead of having no or inadequate knowledge among those who have a

mobile telephone was 1.43 times more likely than those who don't have a mobile telephone. The qualitative study done in Nyanza, Kenya showed that using mobile phone technology enhances linking with health workers, protecting confidentiality, and receiving information and reminders. The same study also concluded that the mobile communications platform holds considerable potential in preventing the MTCT of HIV [44].

Media exposure is expected to be associated with having MTCT of HIV knowledge. Precisely speaking, those women who listen to the radio at most 1 time a week and those who listen to the radio at least 1 time a week were by 24% and 27% more likely to have adequate knowledge of MTCT of HIV instead of not having or having inadequate knowledge of MTCT of HIV respectively. Similarly, those women who were watching television at most 1 time a week and those who were watching television at least 1 time a week were by 28% and 61% more likely to have adequate knowledge of MTCT of HIV instead of not having or having inadequate knowledge of the transmission respectively. This is concordant with the result of the study done in Ethiopia where exposure to mass media was significantly associated with the knowledge of the MTCT of HIV [28] and the cross-sectional study done in the SSA where exposure to mass media had shown a potential effect on HIV-related knowledge [45]. The study from Nigeria also showed that radio and television are the main sources of information to have knowledge of the MTCT of HIV among pregnant women in Nigeria [46]. This might directly be related to the HIV- related educations transmitted through radio and television broadcasting.

Those women who were visited by fieldworkers in the last 12 months before the survey were by 38% more likely to have adequate knowledge of MTCT of HIV instead of not having or having inadequate knowledge of the transmission as compared to those women who were not visited by the fieldworkers in the same period. This finding is consistent with the findings from the cross-sectional study done in Northeast Ethiopia and Tanzania where women who reported receiving information on HIV from health care providers had adequate knowledge of the MTCT of HIV [26, 35]. This finding witness that community-based information, education, and communication of HIV-related information lifts women's knowledge of MTCT of HIV.

## Strength of the study

As the EDHS sampling techniques, data collection techniques, and data processing and management are very strong, the pieces of evidence from the current study are more valid and dependable than the pieces of evidence yielded from the prior small-scall studies done in the country. On the other hand, the current study has sufficient power than other prior small-scale studies done in Ethiopia as the EDHS included a large sample size in the study. The weighting of the data was also done before the analyses to minimizes biases which could have been introduced due to the clustering effect. Moreover, multilevel order logistic modeling was also applied to account for the variation of the level of women's knowledge of the MTCT of HIV across the regions in the country.

## Limitation of the study

The EDHS data, which were extracted and used for the current study were collected using a single-time survey; therefore, the temporal relationship between women's knowledge of the MTCT of HIV and the independent factors identified cannot be ascertained and the yielded evidence should be utilized with cautions. Besides, due to the absence of qualitative data on EDHS, the authors failed to investigate the association between pertinent qualitative variables like socio-cultural factors and women's knowledge of the MTCT of HIV.

## Conclusions

Despite all collective measures put in a place by different stakeholders to prevent the MTCT of HIV in Ethiopia, a large proportion of the Ethiopian women do not know about the transmission of the disease. The current study has also witnessed that there are regional variations in the level of the women's knowledge of the MTCT of HIV. Factors like residing in an urban area, having higher educational status, being from a rich household, owning a mobile phone, listening to radio once a week or more, watching to television once a week or more, and being visited by field workers were associated with having adequate knowledge of the MTCT of HIV. The Ethiopian government along with other world communities has committed to the ambitious plan of ending the HIV epidemic in 2030. But this ambitious plan cannot be realized without promoting the current low level of the Ethiopian women's knowledge of the MTCT of HIV. Therefore, stakeholders working on HIV prevention and control should give due emphasis to promoting mobile phone technology and other media like radio and television by giving due focus to rural residents and poor women to promote the current low level of the knowledge. Emphasis should also be given to the information, education, and communication of the MTCT of the disease through community-based educations.

## Acknowledgments

We are highly thankful to different individuals and organizations that participated in the 2016 EDHS and those who permitted us to access the data sets from the MEASURE DHS website.

## Author Contributions

**Conceptualization:** Mamo Nigatu Gebre.

**Data curation:** Merga Belina Feyasa, Teshome Kabeta Dadi.

**Formal analysis:** Merga Belina Feyasa.

**Methodology:** Mamo Nigatu Gebre, Merga Belina Feyasa, Teshome Kabeta Dadi.

**Software:** Teshome Kabeta Dadi.

**Visualization:** Mamo Nigatu Gebre.

**Writing – original draft:** Mamo Nigatu Gebre, Merga Belina Feyasa, Teshome Kabeta Dadi.

**Writing – review & editing:** Mamo Nigatu Gebre, Merga Belina Feyasa, Teshome Kabeta Dadi.

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
