## [Decision Letter · Decision Letter 0]

1 Apr 2021

PONE-D-20-30425

Levels of Mother-to-Child HIV Transmission Knowledge and Associated Factors among Reproductive-Age Women in Ethiopia: Analysis of 2016 Ethiopian Demographic and Health Survey Data

PLOS ONE

Dear Dr. Gebre,

Thank you for submitting your manuscript to PLOS ONE. After careful consideration, we feel that it has merit but does not fully meet PLOS ONE’s publication criteria as it currently stands. Therefore, we invite you to submit a revised version of the manuscript that addresses the points raised during the review process.

The manuscript has been evaluated by two reviewers, and their comments are available below. You will see the reviewers have commented on the strengths of your manuscript. However, they have also raised a number of concerns that should be addressed before the manuscript can be further considered for publication.

The key concerns noted by the reviewers relate to the description of the study context and the study instruments. Specifically, the reviewers requested clarity regarding the validity of the questionnaire items and scoring procedures in Ethiopia. 

Reviewer 2 has recommended that you cite specific previously published works. As always, we recommend that you please review and evaluate the requested works to determine whether they are relevant and should be cited. It is not a requirement to cite these works. 

Please note that while this manuscript should present new findings, novelty is not a requirement for publication in PLOS ONE: https://journals.plos.org/plosone/s/editorial-and-peer-review-process

We look forward to receiving your revised manuscript.

Kind regards,

Danielle Poole

Staff Editor

PLOS ONE

Journal Requirements:

2. For more information on PLOS ONE's expectations for statistical reporting, please see https://journals.plos.org/plosone/s/submission-guidelines.#loc-statistical-reporting. Please update your Methods and Results sections accordingly.

Reviewers' comments:

Reviewer's Responses to Questions

**Comments to the Author**

1. Is the manuscript technically sound, and do the data support the conclusions?

Reviewer #1: Yes

Reviewer #2: Yes

2. Has the statistical analysis been performed appropriately and rigorously? 

Reviewer #1: Yes

Reviewer #2: Yes

3. Have the authors made all data underlying the findings in their manuscript fully available?

Reviewer #1: No

Reviewer #2: Yes

4. Is the manuscript presented in an intelligible fashion and written in standard English?

Reviewer #1: Yes

Reviewer #2: Yes

5. Review Comments to the Author

Reviewer #1: Overall comment: This is a well written paper with clear results backed by the data. While we understand that the data will be helpful for Ethiopia as a country, the topic in question has studied quite extensively and not much new evidence has been generated. The author will also need to review a few grammatical errors throughout the paper. The discussion of the paper needs some review - it has several statements that are just opinions without backing of literature.

Specific comments:

1. Background: Since this paper focus on PMTCT in Ethiopia, there has been minimal information provided in the background about the PMTCT program in Ethiopia to give context to the reader

2. Background: The administrative set-up of Ethiopia is not highlighted in the background. It is only in the Materials and Methods section that the reader is informed of the study covering the 9 regions and 2 two city administrations in Ethiopia. A sentence or so in the background on the administrative set-up of the country will help the reader contextualize the study coverage.

3. Materials and Methods: Can the author talks briefly about the validation of the questions used to assess clients knowledge in this study. Though used in Tanzania, have these been validated as good measures for MTCT knowledge? Also the categorization of the scoring - has that been validated as well?

4. Discussion: Line 289-292: The author says "The difference might be explained by the differences in the integration of HIV-related education in the maternal health care services in different geographical locations and differences in uptake of maternal health care services among the reproductive-age women residing in the different regions". However, these differences in the integration of HIV related education are not categorically stated and discussed. How are the differences? Do we have literature to demonstrate those differences.

5. Discussion: Line 300-301: The author says: "On the other hand, urban-resident women might have better access to education than rural-residents, which might boost their knowledge of MTCT of HIV". Do we have literature to support this statement?

6. Discussion: Line 321-324: The author says: "This could be explained by the inequalities in accessing educational services, health care services, and social media between women from rich households and those from poor households which might have significant impacts on MTCT of HIV-related knowledge among the women". It will be good to quote the literature to support that there are indeed inequalities between the rich and poor households.

Reviewer #2: Authors wrote an interesting and large paper (15.000 patients) on importatn issue from low income countries. Research from low setting are always important and I suggest to accept the paper on this minor revisions

1. Introduction: update data on burden of HIV globally and in your country. Child with HIV trasmission ogf HIV are defined "children at risk" to worste clincal and social outcome. Please add this concept, see and cite (The At Risk Child Clinic (ARCC): 3 Years of Health Activities in Support of the Most Vulnerable Children in Beira, Mozambique. )

2. Methods and results: are clear and well wrote

3. Discussion: discuss on the nedd of provision of HIV integrated services and compare with other data from Africa (see and citeCapacity assessment for provision of quality sexual reproductive health and HIV-integrated services in Karamoja, Uganda. Afr Health Sci. 2020). Furthermore, on mother and child trasmission give some public health proposal as in other experiece (Pathways of care for HIV infected children in Beira, Mozambique: pre-post intervention study to assess impact of task shifting. BMC Public Health. )

Conclusion: are coherent

References: ref 24 in in Japanese ?

6. PLOS authors have the option to publish the peer review history of their article (what does this mean?). If published, this will include your full peer review and any attached files.

Reviewer #1: **Yes: **Dr Caspian Chouraya

Reviewer #2: **Yes: **Francesco Di Gennaro

---

## [Author Response · Author response to Decision Letter 0]

9 May 2021

PONE-D-20-30425

Levels of Mother-to-Child HIV Transmission Knowledge and Associated Factors among Reproductive-Age Women in Ethiopia: Analysis of 2016 Ethiopian Demographic and Health Survey Data

PLOS ONE

Authors’ response: We are immensely grateful to both the reviewers and the editorial team for their invaluable constructive comments which helped us a lot to revise our manuscript. 

A point-by-point response to reviewers’ comments 

Reviewer #1: Overall comment: This is a well written paper with clear results backed by the data. While we understand that the data will be helpful for Ethiopia as a country, the topic in question has studied quite extensively and not much new evidence has been generated. The author will also need to review a few grammatical errors throughout the paper. The discussion of the paper needs some review - it has several statements that are just opinions without backing of literature.

# Response: We have extensively edited the grammar and revised the discussion

Specific comments:

1. Background: Since this paper focus on PMTCT in Ethiopia, there has been minimal information provided in the background about the PMTCT program in Ethiopia to give context to the reader

#Response: Now we have provided detailed account of information on PMTCT provision in Ethiopia on the page 3, line 113-121 of the manuscript. 

2. Background: The administrative set-up of Ethiopia is not highlighted in the background. It is only in the Materials and Methods section that the reader is informed of the study covering the 9 regions and 2 two city administrations in Ethiopia. A sentence or so in the background on the administrative set-up of the country will help the reader contextualize the study coverage.

#Response: Now we have provided a highlight information about Ethiopia in the background of the manuscript on page 4, line 133-140

3. Materials and Methods: Can the author talks briefly about the validation of the questions used to assess clients knowledge in this study. Though used in Tanzania, have these been validated as good measures for MTCT knowledge? Also the categorization of the scoring - has that been validated as well?

#Response: In the current study we used the 2016 EDHS data. For collecting the EDHS data, standard protocols and three types of tools; the Household Questionnaire, the Woman’s Questionnaire, and the Man’s Questionnaire were used. Further contextualization and standardization of the questionnaires were also done by governmental and non-governmental shareholders to maintain the validity of the tools. We have described this in the method part of the manuscript on page 5, line 154-157. We used the study done in Tanzania as an additional reference for the logical operationalization of the women’s knowledge of the mother-to-child transmission of HIV. 

4. Discussion: Line 289-292: The author says "The difference might be explained by the differences in the integration of HIV-related education in the maternal health care services in different geographical locations and differences in uptake of maternal health care services among the reproductive-age women residing in the different regions". However, these differences in the integration of HIV related education are not categorically stated and discussed. How are the differences? Do we have literature to demonstrate those differences.

#Response: we have edited this as “The difference might be explained by the differences in uptake of maternal health care services among the reproductive-age women residing in the different regions,” and supported the implication with literature on page 11, line 331-339. 

5. Discussion: Line 300-301: The author says: "On the other hand, urban-resident women might have better access to education than rural-residents, which might boost their knowledge of MTCT of HIV". Do we have literature to support this statement?

 #Response: we have revised this as “This might be because urban-resident women might have better access to maternal health care services and mass media than rural-resident women,” and supported the implication with literature on 11, line 345-349. 

6. Discussion: Line 321-324: The author says: "This could be explained by the inequalities in accessing educational services, health care services, and social media between women from rich households and those from poor households which might have significant impacts on MTCT of HIV-related knowledge among the women". It will be good to quote the literature to support that there are indeed inequalities between the rich and poor households.

#Response: We have quoted literature for the implication on page 12, line 378-384. 

Reviewer #2: 

1. Introduction: update data on burden of HIV globally and in your country. Child with HIV trasmission ogf HIV are defined "children at risk" to worste clincal and social outcome. Please add this concept, see and cite (The At Risk Child Clinic (ARCC): 3 Years of Health Activities in Support of the Most Vulnerable Children in Beira, Mozambique. )

#Response: we have updated the information on the burden of the HIV from the global perspective to the study area on page 1, line 61-70. We have also included a statement about a “child at risk” in a context of this study and quoted references for it. 

2. Discussion: discuss on the nedd of provision of HIV integrated services and compare with other data from Africa (see and citeCapacity assessment for provision of quality sexual reproductive health and HIV-integrated services in Karamoja, Uganda. Afr Health Sci. 2020). Furthermore, on mother and child trasmission give some public health proposal as in other experiece (Pathways of care for HIV infected children in Beira, Mozambique: pre-post intervention study to assess impact of task shifting. BMC Public Health.)

#Response: In the current study, we have not categorically studied the provision of the HIV integrated services; therefore, we could not compare our findings with the findings from the reference suggested. The study done in Mozambique titled with “Pathways of care for HIV infected children in Beira, Mozambique: pre-post intervention study to assess impact of task shifting,” was to evaluate the effectiveness of task-shifting (TS) from clinical officers to maternal and child nurses to improve care for HIV positive children < 5 years old. The study concluded that the task-shifting was effective in caring for HIV positive children. However, the findings of the study were not related to the women’s knowledge of the MTCT of the HIV infection. Therefore, we did not used it as a reference. 

References: ref 24 in in Japanese ?

#Response: Since we have added another references in the background parts of the manuscript, the sequential order of the reference 24 became reference 36. it is an editorial error; we have corrected it.

---

## [Decision Letter · Decision Letter 1]

9 Aug 2021

Levels of Mother-to-Child HIV Transmission Knowledge and Associated Factors among Reproductive-Age Women in Ethiopia: Analysis of 2016 Ethiopian Demographic and Health Survey Data

PONE-D-20-30425R1

Dear Dr. Gebre,

We’re pleased to inform you that your manuscript has been judged scientifically suitable for publication and will be formally accepted for publication once it meets all outstanding technical requirements.

Kind regards,

Avanti Dey, PhD

Staff Editor

PLOS ONE

Additional Editor Comments (optional):

This manuscript is now ready for acceptance. One minor point has been raised by Reviewer #1, so please ensure that HAART  is replaced with ART throughout the manuscript.

Reviewers' comments:

Reviewer's Responses to Questions

**Comments to the Author**

1. If the authors have adequately addressed your comments raised in a previous round of review and you feel that this manuscript is now acceptable for publication, you may indicate that here to bypass the “Comments to the Author” section, enter your conflict of interest statement in the “Confidential to Editor” section, and submit your "Accept" recommendation.

Reviewer #1: All comments have been addressed

Reviewer #2: All comments have been addressed

2. Is the manuscript technically sound, and do the data support the conclusions?

Reviewer #1: Partly

Reviewer #2: Yes

3. Has the statistical analysis been performed appropriately and rigorously? 

Reviewer #1: Yes

Reviewer #2: Yes

4. Have the authors made all data underlying the findings in their manuscript fully available?

Reviewer #1: No

Reviewer #2: Yes

5. Is the manuscript presented in an intelligible fashion and written in standard English?

Reviewer #1: Yes

Reviewer #2: Yes

6. Review Comments to the Author

Reviewer #1: The comments have been adequately addressed. However, please note that HAART is now a redundant term and rarely used. Please replace with ART throughout the manuscript

Reviewer #2: cogratulations. I think the paper can be now accept. The paper is interesting and also the setting of research

7. PLOS authors have the option to publish the peer review history of their article (what does this mean?). If published, this will include your full peer review and any attached files.

Reviewer #1: **Yes: **Dr Caspian Chouraya

Reviewer #2: No

---

## [Editor Report · Acceptance letter]

11 Aug 2021

PONE-D-20-30425R1 

Levels of Mother-to-Child HIV Transmission Knowledge and Associated Factors among Reproductive-Age Women in Ethiopia: Analysis of 2016 Ethiopian Demographic and Health Survey Data 

Dear Dr. Gebre:

I'm pleased to inform you that your manuscript has been deemed suitable for publication in PLOS ONE. Congratulations! Your manuscript is now with our production department. 

Kind regards, 

on behalf of

Dr. Avanti Dey 

Staff Editor

PLOS ONE